# Protein Functional Effector (pfe) Noncoding RNAS Are Identical to Fragments from Various Noncoding RNAs

**DOI:** 10.3390/ijms26146870

**Published:** 2025-07-17

**Authors:** Roberto Patarca, William A. Haseltine

**Affiliations:** 1ACCESS Health International, 384 West Lane, Ridgefield, CT 06877, USA; william.haseltine@accessh.org; 2Feinstein Institutes for Medical Research, 350 Community Dr, Manhasset, NY 11030, USA

**Keywords:** protein functional effector RNA, noncoding RNA, RNA-protein interactions, transfer RNA fragment, ribosomal RNA fragment, microRNA fragments, Y RNA fragments, PD-1/PD-L1 interaction, cancer, RNA modifications

## Abstract

Protein functional effector (pfe)RNAs were introduced in 2015 as PIWI-interacting-like small noncoding (nc)RNAs and were later categorized as a novel group based on being 2′-O-methylated at their 3′-end, directly binding and affecting protein function, but not levels, and not matching known RNAs. Here, we document that human pfeRNAs match fragments of GenBank database-annotated human ncRNAs. PDLpfeRNAa matches the 3′-half fragment of a mitochondrial transfer (t)RNA, and PDLpfeRNAb matches a 28S ribosomal (r)RNA fragment. These PDLpfeRNAs are known to bind to tumor programmed death ligand (PD-L)1, enhancing or inhibiting its interaction with lymphocyte PD-1 and consequently tumor immune escape, respectively. In a validated 8-pfeRNA-set classifier for pulmonary nodule presence and benign vs. malignant nature, seven here match one or more of the following: transfer, micro, Y, PIWI, long (lnc)RNAs, and a PDLpfeRNAa fragment. The previously identified chromosomal locations of these pfeRNAs and their matches partially overlap. Another 2-pfeRNA set was previously determined to distinguish between controls, patients with pulmonary tuberculosis, and those with lung cancer. One pfeRNA, previously shown to bind p60-DMAD and affect apoptosis, complements small nucleolar RNA SNORD45C, matching smaller 18S rRNA and lncRNA segments. Thus, pfeRNAs appear to have a common origin with known multifunctional ncRNA fragments. Differential modification may contribute to the multifunctionality of ncRNAs. For instance, for tRNA fragments, stabilizing 3′-end 2′-O-methylation, 3′-aminoacylation, and glycosylation modifications may regulate protein function, translation, and extracellular effects, respectively. One ncRNA gene can encode multiple fragments, multiple genes can encode the same fragment, and differentially modified ncRNA fragments might synergize or antagonize each other.

## 1. Introduction

The eukaryotic transcriptome primarily consists of noncoding (nc)RNAs, which predominantly underlie biological diversity and regulation [1,2]. Due to their abundance, structural flexibility, and varied conformations, RNAs can regulate nearly every aspect of cellular physiology, influencing chromatin organization, gene expression regulation, biochemical reaction catalysis, cellular signaling pathways, embryogenesis, and immune responses, among other processes [1,3,4].

ncRNAs can be broken down into fragments that are not simply byproducts of nonspecific degradation or remnants of RNA precursor maturation but result from specific enzymatic pathways and have distinctive regulatory functions [5]. Even transfer (t)RNAs and ribosomal (r)RNAs, considered housekeeping RNAs, and their fragments have regulatory functions [5,6,7,8,9,10,11].

Among ncRNAs, protein functional effector (pfe)RNAs were characterized as novel ones that play a critical role in tumorigenesis and the differentiation of non-small cell lung cancer (NSCLC) [12,13,14,15]. These 26–60 nucleotide-long pfeRNAs exhibit distinct features, including 2′-O-methylation at the 3′ end, and direct interactions with target proteins rather than transcripts, thereby regulating protein function without altering their levels [16]. All pfeRNAs thus far characterized have been reported not to match other known RNAs.

Two pfeRNAs, termed PDLpfeRNAs, affect the interaction between the programmed death (PD)-1 surface protein on T lymphocytes and the PD-1 ligand (PD-L1) on tumor cells. By binding to PD-L1, PDLpfeRNAa enhances the PD-1/PD-L1 interaction, while PDLpfRNAb inhibits it, favoring or inhibiting, respectively, tumor immune escape [16,17]. Whether PDLpfeRNAs’ structural features or differential binding affinities to the PD-L1/PD-1 complex mediate their effects on PD-1/PD-L1 interactions remains undetermined.

Normal tissues show a balanced expression of PDLpfeRNA a and b, while PDLpfeRNAa expression is significantly higher during tumorigenesis [17]. In patients with unresectable malignant pleural mesothelioma, the plasma relative expression levels of these pfeRNAs predicted response to first-line treatment [17]. Said treatment included the PD-L1 inhibitor dorvalumab, cisplatin, which synergizes with PD-1/PD-L1 inhibition, and pemetrexed, which induces transcriptional activation of PD-L1 and secretion of cytokines that further increase PD-L1 levels. PDLpfeRNAa/PDLRNAb plasma relative expression levels had significant prognostic value for overall and progression-free survival, regardless of histological subtype and age [17].

Another eight pfeRNAs, termed pfeRNA a-h, were validated as a set in a multicenter study in China and the United States as cost-effective, non-invasive, liquid biopsy biomarkers for the presence of pulmonary nodules and their nature [18]. A classifier based on these pfeRNAs differentiated between the presence and absence of pulmonary nodules (mean 96.2% specificity, 97.35% specificity), and malignant vs. benign pulmonary nodules (mean 77.1% sensitivity, 74.25% specificity) [18].

Lastly, two additional pfeRNAs, termed PIWI-interacting-like small RNAs (piR-L)-163 [12] and piR-L-138 [15] were characterized. piR-L-163 directly binds to phosphorylated ERM (*p*-ERM) proteins, playing a critical role in ERM activation and the regulation of signal transduction. Cell cortex ERM (ezrin, radixin, and moesin) proteins bridge transmembrane and cytoskeleton proteins. The phosphorylation of ERM proteins exposes binding sites for transmembrane proteins, such as EBP50, and the cytoskeleton, including filamentous actin. piR-163 is expressed in immortalized human bronchial epithelial cells, with distinctive downregulation of expression in lung cancer cells. The second pfeRNA, piR-L-138, is upregulated in association with apoptosis inhibition in resistance to cisplatin-based chemotherapy in lung squamous cell carcinoma (LSCC) cells and patient-derived xenograft LSCC models [15]. piR-L-138, directly interacts with the phosphorylated mouse double minute 2 homolog (p60-MDM2) in response to cisplatin-based therapy of LSCC, inducing chemoresistance by inhibiting apoptosis [15].

Here, comparisons of pfeRNA primary sequences against sequences in the GenBank database revealed that the pfeRNAs mentioned above match fragments derived from various annotated ncRNAs, including mitochondrial and nuclear transfer, ribosomal, micro, Y, PIWI-interacting, and long ncRNAs. Two pfeRNAs are segments of other pfeRNAs; sometimes, pfeRNAs match more than one ncRNA type; and one pfeRNA had no matches in the GenBank database. piR-L-163 matched an intronic region in the human laminin subunit gamma 2 oncogene. piR-L138 matched complementary sequences in small nucleolar and long ncRNAs, and shorter sense regions in 18S rRNA and long ncRNA fragments. These findings underscore that one ncRNA-type gene can express more than one fragment, and more than one ncRNA-type gene can give rise to the same fragment. Furthermore, differential modifications to each RNA fragment, determining their stability level and specific function, may increase the versatility of this regulatory network. The structural bases of pfeRNAs, which direct interactions with proteins and regulate protein function without altering protein levels, remain to be determined beyond what has been covered in the literature.

## 2. Results

### 2.1. Analysis of pfeRNAs Related to PD-1/PD-L1 Interactions, Termed PDLpfeRNAs

PDLpfeRNAa, induced during non-small cell lung cancer tumorigenesis, binds to PD-L1 and enhances PD-1/PD-L1 interaction, favoring tumor immune escape [16,17]. A search of the GenBank database using the 39-nt-long human PDLpfeRNAa as query sequence revealed that it is identical to the 3′-half fragment of the human mitochondrial tRNA for glutamic acid (Glu) (Figure 1A) [19]. This 3′-half fragment could be generated via the action of angiogenin at the level of the tRNA’s anticodon loop [6,20] (Figure 1B). The visualized secondary structure (Figure 1B) has an estimated free minimal energy (ΔG) of −6.8 Kcal/mol.

A similar analysis revealed that the human 41-nucleotide-long PDLpfeRNAb, which is also induced during tumorigenesis, is identical to a fragment towards the 3′ end in human 28S ribosomal (rRNA), part of its domain 5 (Figure 2A,B). The visualized secondary structure of this rRNA fragment has an estimated ΔG of −7.4 Kcal/mol (Figure 2B). Like PDLpfeRNAa, PDLpfeRNAb binds to PD-L1 [6,17]; however, in contrast to PDLpfeRNAa, it inhibits the functional interaction between PD-1 and PD-L1, consequently inhibiting tumor immune escape, as do the PD-1 inhibitors in cancer immunotherapy (Figure 2B).

### 2.2. Analysis of Plasma pfeRNAs Related to Pulmonary Nodules: pfeRNAa-h

A set of eight pfeRNAs was characterized and validated as a low-cost, noninvasive liquid-biopsy classifier for pulmonary nodule presence vs. absence, and nature, i.e., benign vs. malignant [18]. A search of the GenBank database revealed identical sequences in fragments of Y RNA, microRNA-21, Piwi-interacting and long ncRNAs, and mitochondrial and nuclear tRNAs, including one, pfeRNAb, that is a fragment of PDLfeRNAa and therefore also matching the mitochondrial tRNA Glu 3′-half (Figure 3). One of the eight pfeRNAs, pfeRNAh, did not have any highly similar matches in the GenBank database (Figure 3).

GenBank accession numbers for the matches are: Ro60 Y4 RNA (NR_004393.1) [21], microRNA21 (NR_029493.1) [22], Hy40 Ro RNA (L32608.1) [23], tRNA-Lys (HG983908.1, HG983910.1, HG983915.1, HG983916.1) [24,25,26]; piRs: piR-31104 (DQ57099.2), piR-3452 (DQ597346.1), piR-35410 (DQ597344.1), piR-35411(DQ597345.1), piR-35412 (DQ597346.1), piR-35413 (DQ597347.1), piR-35413 (DQ597347.1), PiR-143604 (DQ575492.1) [27]; lncRNA 1128 (MN298243.1); and mt tRNA Val (LC530724.1) [19].

Chromosomal locations of pfeRNAa-h were predicted in the publication that characterized pfeRNAs a-h using the QIAGEN CLC Genomics Workbench 10.11 software package [18]. Chromosomal locations of the matched sequences in the annotated GenBank database, as determined here, overlap with those of 5 of the 7 pfeRNAs (Figure 4). For the two pfeRNAs without chromosomal location overlaps, one, pfeRNAb, matches PDLpfeRNa. We infer it has a mitochondrial genome localization based on its identical sequence with mitochondrial tRNA Glu. The other, pfeRNAc, is identical to a fragment spanning 48 nucleotides in microRNA-21 whose gene has been mapped to human chromosome 17q3.2 in the 11th intron of the gene encoding the TMEM49 transmembrane protein, which is a precursor of VMP1 (vacuole membrane protein (VMP)1 (Figure 4) [28,29]. This contrasts with the location on chromosome 5 in the reference characterizing pfeRNAc and using a software package for location assignment [18]. The chromosomal locations of these pfeRNAs need to be determined not through prediction, but rather through a more direct methodology, as is the case for the annotated sequences to which they match here.

### 2.3. Analysis of Plasma pfeRNAs Related to p-ERM Proteins and p60-MDM2

The pfeRNA, piR-L-163, that binds to *p*-ERM proteins (ERM-pfeRNA) affecting signal transduction matched a sequence in the intron after exon 10 of the human laminin subunit gamma 2 oncogene (Figure 5A). The pfeRNA, piR-L-138, that binds to p60-MDM2 (p60-MDM2-pfeRNA) influencing apoptosis matched the complementary sequences of a fragment of the small nucleolar RNA SNORD45C and a smaller segment of a long noncoding RNA (in green in Figure 5B). p60-MDM2-pfeRNA also matches a similarly smaller segment of 18S rRNA and a lncRNA (Figure 5B).

GenBank accession numbers for matches to p60-MDM2-pfeRNA are (Figure 5): small nucleolar RNA C/D box 45 C (1-79: LN848108.1, 1-78: NR_003042.1, 1-83: LN848102.1, 1-84: NR_002749.1); LNC_000006 lncRNA (MN308643.1); 18S rRNA, chain 2 (8UKB_52, 8ZDC-2, 6ZOJ_2, 6ZON_2,5A2Q_2, 6G4S_2), 45S pre-ribosomal N2 rRNA (NR_146144.1, NR_146117.1), 18S rRNA chain L1 (1-1,872: 7MQ9_L1), 18S rRNA chain B1 (1-1,1869: 5AJO_B1), 18S pre-ribosomal 4 (NR_146119.1); and LNC_00012 (MN308649.1).

## 3. Discussion

The present study identified matches between structurally and functionally characterized pfeRNAs and segments of annotated ncRNA sequences in the GenBank database, further exemplifying the known ability of one ncRNA to encode multiple fragments and highlighting that the same fragment can originate from multiple ncRNAs. The relevance of these matches is further underscored by the overlap between the published chromosomal locations for 5 of 7 pulmonary nodule-related pfeRNAs, determined using a genomic software program, and those annotated in the GenBank database matches. Moreover, matching sequences may exhibit functional overlaps, as observed in studies on similar fragments. Only one of the 12 characterized pfeRNAs had no matches in the GenBank database. A pfeRNA that interacts with the epidermal growth factor receptor (EGFR) and affects cell growth and colony formation in lung cancer H226 cells was mentioned in a publication [16]. However, its sequence was not provided and therefore not included in this analysis.

### 3.1. PD-L1-Binding pfeRNAs (PDLpfeRNAs a and b)

**PDLpfeRNAa**, which was previously shown to bind to the PD-L1 protein and enhance PD-1/PD-L1 interaction, thereby favoring tumor immune escape [16,17], is identical in sequence to the 3′-half fragment of human mitochondrial tRNA Glu here. tRNA fragments of different types and lengths are known to regulate cell viability, differentiation, and homeostasis in both health and disease, including cancer, through diverse mechanisms in various processes, such as ribosome biogenesis and gene silencing, by binding to complementary nucleic acids and various proteins [30,31,32,33]. tRNA halves bind to specific proteins. For instance, a 5′-tRNA half binds to the YBX1 (YB-1) protein, promoting stress granule formation and inhibiting global protein translation during stress conditions [6]. Ribosomes with a vacant A site, whose abundance increases during cellular stress, activate angiogenin [34], which cleaves at the anticodon loop, generating tRNA halves. The match described here makes it possible that the 3′-half fragment of human mitochondrial tRNA Glu is the origin of PDLpfeRNA or, like it, binds to PD-L1.

**PDLpfeRNAb**, which was also previously shown to bind to PD-L1 but inhibits PD-1/PD-L1 interaction and tumor immune escape [16,17], is identical to a sequence towards the 3′-end of human 28S rRNA here. rRNA fragments can associate with specific proteins, including those binding small RNAs, such as the P19 protein [35]. Consistent with the match shown here between PDLpfeRNAb and a 28S rRNA fragment, the generation of 28S rRNA fragments in tumor cells has been linked to cell death and cytotoxicity, suggesting that these fragments may serve as markers or mediators of cancer cell stress responses [36].

28S rRNA fragments dominate the rRNA fragment pool [37]. The primary precursor, 45S pre-rRNA, undergoes sequential cleavage to generate intermediate transcripts (32S or 36S pre-rRNAs) before maturing into 28S rRNA. During this process, parallel pathways involving endonucleases and exonucleases generate rRNA fragments [38]. From ticks to humans, rRNA fragments, whose length progresses with one nucleotide difference, align to the 5′- and 3′-ends of the 5.8S and 28S rRNA genes and are predominantly expressed over those in the body of the rRNA genes [5].

### 3.2. Pulmonary Nodule-Related pfeRNAs

A set of eight pfeRNAs, pfeRNA a-h, has been shown to classify presence vs. absence and benign vs. malignant pulmonary nodules [18].

**pfeRNAa** here matches a fragment spanning almost the entire 3′-half of Ro60 Y4 RNA. The ring-shaped Ro60 protein is known to form a ribonucleoprotein complex by binding RNAs such as Y RNAs, which regulate the subcellular localization of Ro60, tether it to effector proteins, and regulate other RNAs’ access to its central cavity [21,23].

Because mammalian cells and bacteria lacking Ro60 are sensitized to ultraviolet irradiation, Ro60 function in RNA quality control may be particularly relevant during exposure to some environmental stressors, where Y RNA and its expression increase. Y RNA fragments are dysregulated in tumors and associated with cancer progression [39,40]. Ro60 RNP is a target of autoantibodies in patients with some rheumatic diseases, potentially contributing to their initiation and progression (reviewed in [21]).

**pfeRNAb** matches, except for five nucleotides, PDLpfeRNAa, which, in turn, matches the 3′-half fragment of mitochondrial tRNA Glu. The predicted chromosomal location for pfeRNAb was on chromosomes 13 and X in the publication that characterized pfeRNAb [18]. Although a study identified nuclear genomic sequences resembling mitochondrial tRNAs, with Glu tRNAs being among the most represented [41], the discrepancy in predicted chromosomal location between PDLpfeRNAa, based on its similarity to mitochondrial tRNA Glu, and that for pfeRNAb, based on the QIAGEN CLC Genomics Workbench 10.1.1 [18], remains to be elucidated.

**pfeRNAc** here matches a 48-nucleotide segment of the 72-nucleotide-long human microRNA (miR)-21.


What is known about miRs and, in particular, miR-21?


miRs silence gene expression by binding partially complementary sequences within target messenger RNAs. miRNAs recognize canonical target sites by base-pairing in their 5′ region. However, there are non-canonical target sites [42].

First discovered in 2008 in human stem cells [43], isomicroRNAs (isomiRs) are miR sequence variants that differ from the canonical miR sequence by changes, such as trimming at the 5′ end (5′-isomiRs), often due to alternative cleavage during mRNA processing; trimming at the 3′ end; nucleotide additions at the 3′ end, often through post-transcriptional modifications; internal substitutions; or combinations of the above including trimming at both 5′ and 3′ ends (mixed-type) [44,45].

5′- and 3′-isomiRs are widespread and represent approximately half of miR copies in cells and tissues [46]. IsomiRs are generated through regulated processes during miRNA biogenesis and are often conserved across species [47,48]. IsomiR expression changes in diseases such as cancer, making them potential biomarkers [46,47,49], as demonstrated for PNpfeRNAc [18]. pfeRNAc lacks the first seven nucleotides at the 5′ end (most of the seed sequence from nucleotides 2 to 8 that binds to the target messenger RNA) and 17 nucleotides at the 3′ end, rendering it a mixed-type isomiR. pfeRNAc would be a templated isomiR because its sequence matches the parental miR21 gene [50].

miR-21 is classified as an “oncomiR,” a bona fide oncogene consistently upregulated in nearly all types of cancers. It promotes cell proliferation, migration, invasion, and survival by targeting tumor suppressor genes such as PTEN and PDCD4 [51,52,53]. Other associations of mR-21 include cardiac and pulmonary fibrosis [51], immunity, inflammation [52], and osteogenesis [54].

miR-21 acts as an oncogene by targeting tumor suppressor genes and promoting cell proliferation, migration, and invasion in lung cancer cells. In non-small cell lung cancer (NSCLC), miR-21 is consistently overexpressed and detectable in both serum and sputum. Elevated miR-21 levels are associated with poor prognosis, advanced clinical stage, lymph node metastasis, and lower survival rates [55].

Cells have multiple isomiRs of miR-21 [56]. Overall, for 5′ isomiRs, changes in the miR 5′ seed region (nucleotides 2–8), as present in PNpfeRNAc, can alter the set of mRNA targets (targetome shifting), potentially leading to different regulatory outcomes. IsomiRs may also act with canonical miRNAs to regulate the same pathways or have distinct, even opposing, effects (cooperative or divergent function). For instance, a 3′ isomiR of miR-21 suppresses hepatoma cell growth [57]. Moreover, consistent with the association of pfeRNAc with pulmonary nodules and their presence vs. absence and benign vs. malignant nature, miR-21 fragments drive tumor progression by silencing tumor suppressors like PTEN, promoting uncontrolled cell proliferation in lung and hepatocellular carcinomas [58].

**pfeRNAd** overlaps except for 3 nucleotides with **pfeRNAg**, which matches almost the entire 5′-half of Y4 RNA, whose 3′-half matches pfeRNAa. Therefore, the items discussed for Y4 RNA apply to pfeRNAs d and g.

**pfeRNAe** here matches, except for its first five nucleotides, a 5′ fragment of tRNA Lys, and minus an additional nucleotide, several piRs and a lncRNA. piRs participate in transposon silencing, heterochromatin modification, germ cell maintenance, and tumorigenesis [15,59,60,61]. These matches illustrate how fragments can differ by as little as one nucleotide, as has been described for rRNA fragment series [5].

**pfeRNAf** matches mitochondrial tRNA Val except for the latter’s first 19 nucleotides. Mitochondrial-derived tRNA fragments are biomarkers for chronic lymphocytic leukemia and Mycobacterium tuberculosis infection [58].

**pfeRNAh** had no highly similar matches with GenBank sequences.

### 3.3. ERM-pfeRNA and p-60-MDM2-pfeRNA

A set of two pfeRNAs, one binding *p*-ERM proteins (ERM-pfeRNA) influencing signal transduction [12], and the other p60-MDM2 (*p*-60-MDM2 pfeRNA) affecting cancer cell apoptosis and chemoresistance to cisplatin-based therapy [15], distinguishes among controls and pulmonary tuberculosis and lung cancer patients. Likewise, Gu et al. [62] later developed a TRY-RNA signature composed of tRNA fragments, rRNA-derived small RNAs, and YRNA-derived small RNAs from human peripheral blood mononuclear cells, which exhibited diagnostic potential for precise discrimination between healthy controls, lung cancer and pulmonary tuberculosis [62]. In the latter study, tRNALys-derived small RNAs (in this report one matched with pulmonary nodule-related pfeRNAe), along with fragments from tRNA-Ala, tRNA-Asn, tRNA-Leu, and tRNA-Tyr, were the only five tRNA-derived small RNA groups that were upregulated in the lung cancer patients relative to controls and pulmonary tuberculosis patients [62].

As Gu et al. [62] concluded, changes in the composition of tRNA-derived, rRNA-derived, and Y RNA-derived small RNA may result in altered ribosome heterogeneity, which directs the cell to a specific functional state [63]. Permutations of various small noncoding RNAs may provide specificity to distinguish complex diseases and represent a “disease RNA code” in lung cancer screening [62].

**ERM-pfeRNA (piR-L-163)** here matches an intron segment after exon 10 in the laminin subunit gamma 2 (LAMC2) oncogene. Laminins, a family of extracellular matrix glycoproteins, are the principal noncollagenous constituent of basement membranes. In certain cancers, notably intrahepatic cholangiocarcinoma, LAMC2 facilitates tumor growth and metastasis through molecular pathways including epidermal growth factor receptor (EGFR) signaling. LAMC2 constitutes a diagnostic and prognostic biomarker in neoplasias [64,65,66,67,68,69,70,71]. There is no published evidence that introns in said gene encode ncRNAs. However, through further processing after splicing, introns in other genes encode small regulatory RNAs, such as miRs, small nucleolar RNAs, long intronic RNAs, and circular RNAs [72,73,74].

***p*-60-MDM2-pfe RNA (piR-L-138)** here is complementary to a segment of the small nucleolar RNA SNORD45C and a smaller segment of an lncRNA, which may antagonize it. *p*-60-MDM2-pfeRNA matches a similar segment of 18S rRNA and another lncRNA.


What is known about small nucleolar RNAs?


Some fragments originating from small nucleolar RNAs function like miRNAs [75,76,77]. Dysregulation of small nucleolar RNAs plays a vital role in lung tumorigenesis, and sputum small nucleolar RNA biomarkers might improve lung cancer diagnosis [78]. The rates of nucleolar ribosome production and ribosomal protein biosynthesis are tightly correlated with cell growth and proliferation rates. Deregulation of factors, including oncogenes, controlling these processes, especially ribosome biosynthesis, can lead to cell transformation [79].

A study characterized 21 antisense small nucleolar RNAs from human cells required for site-specific 2′-O-methylation of preribosomal RNA through direct base pairing interactions [80]. The antisense element and the small nucleolar RNA’s D or D’ box provide the information necessary to select the target nucleotide for the methyltransfer reaction [80,81,82]. Most small nucleolar RNAs modify rRNA [83]. This is consistent with the matching here of *p*-60-MDM2 pfeRNA with an antisense small nucleolar RNA fragment and a sense 18S rRNA fragment, and the effects of this pfeRNA on cancer cell apoptosis and chemoresistance to cisplatin-based therapy [15]. Small nucleolar RNAs regulate alternative splicing through sequence-specific RNA interactions and the formation of protein complexes [83].

### 3.4. pfeRNAs, glycoRNAs, and Nicked tRNA Halves Among RNAs Affecting Cell Surface and Extracellular Protein Functions and PD-1/PD-L1 Interactions

pfeRNAs are not the only small ncRNAs that have an extracellular effect. For instance, as with proteins and lipids, glycosylation modifies RNAs, giving rise to glycoRNAs primarily present at the cell surface. GlycoRNAs consist of small nuclear RNAs modified with secretory N-glycans rich in sialic acid and fucose via their attachment to the modified base 3-(3-amino-3-carboxypropyl) uridine (acp3U) [84,85]. GlycoRNAs in mammals and other eukaryotes interact with antibodies and cellular receptors, influencing neutrophil recruitment, immunity, and pathogenesis [84,85,86,87,88,89]. GlycoRNAs, including glycosylated transfer, ribosomal, small nuclear, small nucleolar, and Y noncoding RNAs, could explain why RNAs, traditionally considered intracellular molecules, act as autoantigens [86]. This is beyond the attachment of RNAs to autoantigenic proteins, such as Ro, as described above.

Here, we showed that pfeRNAs thus far characterized and with an available sequence in the literature are identical or highly similar to fragments of transfer, ribosomal, Y, and small nucleolar noncoding RNAs, which might become extracellular after being glycosylated. For instance, Y RNAs are classically mainly cytoplasmic with a minor fraction in the nucleus [90], and tRNAs localize to the soluble cytosol and nucleus. The biogenesis of sialylated glycans occurs across many subcellular compartments, including the cytosol (processing of ManNAc to Neu5Ac), the nucleus (charging of Neu5Ac with CMP), and the secretory pathway (where sialyltransferases add sialic acid to the termini of glycans) [84].

As mentioned above, antibodies targeting RNA have been associated with systemic lupus erythematosus [91]. Besides the described role of RNA in PD-1/PD-L1 interactions, another example of RNA-mediated improvement of immune checkpoint blockade treatment of glioblastomas in mice is the lupus-derived 4H2 anti-guanosine autoantibody, which enters cells through a membrane transit nucleoside salvage-linked pathway after systemic administration. It then binds endogenous RNA, stimulating the cytoplasmic pattern recognition receptor cyclic GMP-AMP synthase (cGAS), immune signaling, and cytotoxicity [92].

Fragments of specific long-coding and noncoding RNAs are present on the surface of cells, indicating an expanded role for RNA in cell–cell and cell–environment interactions [93]. tRNA and rRNA fragments, Y and microRNAs, are present in extracellular vesicles [94], and there is non-vesicle-associated extracellular RNA [95]. Among the latter, full-length tRNAs containing broken phosphodiester bonds (i.e., nicked tRNAs) are stable reservoirs of tRNA halves in cells and biofluids [96]. A recently developed protocol distinguishes between structurally distinct but sequence-identical tRNA-derived fragments and nicked tRNAs, disentangling their biological functions [97].

### 3.5. 3′-end 2′-O-Methylation Is Also Present in ncRNAs Other than the Characterized pfeRNAs

Methylation of the 2′-hydroxyl group of the ribose sugar of a nucleotide, or 2′-O-methylation, is among the most common RNA modifications across kingdoms of life, increasing RNA stability. It is a characteristic feature of the 3′-end of Piwi-interacting RNAs in animals and miRNAs in plants carried out by the S-adenosylmethionine-dependent methyltransferase (MTase) Hen1 [98,99,100,101]. In the ciliated protozoan Tetrahymena, 3′-end 2′-O-methylation on a selected class of small RNAs regulates the function of a specific RNA interference pathway [102].

2′-O-methylation is also a highly abundant posttranscriptional modification at internal sites in ncRNAs, such as ribosomal, transfer, small nuclear RNAs [103,104,105,106,107]. Each methylation site in tRNA, rRNA, or other RNAs is typically modified by a distinct methyltransferase, highlighting the specificity of these enzymes for their RNA substrates and target sites [108]. 2′-O-methylation is an essential feature of the 5′ cap of eukaryotic mRNAs [109].

In plants, specific 5′-tRNA fragments (e.g., tRF-5a Ala and tRF-5b Gly) bear 3′-end-2′-O-methylation, which protects them from degradation and may guide Argonaute proteins for gene regulation. This modification is independent of HEN1, the enzyme responsible for miRNA methylation, and is conserved across species [110]. It remains to be determined if the 3′-ends of fragments from human transfer (mitochondrial and nuclear), ribosomal, Y, and micro RNAs undergo 2′-O methylation as characterized for the pfeRNAs as a distinguishing feature, along with binding to proteins to influence their function directly.

Other chemical groups might be present in some of the ncRNAs instead of the 2′-O-methyl group and might be removed and substituted for it. For instance, tRNA 3′-halves are aminoacylated, as are the full tRNAs from which they originated [111]. However, 2′-O-methylation is not universally required for RNA-protein binding, and tRNA and rRNA fragments can retain 2′-O-methylated nucleotides depending on the cleavage site [108,112].

## 4. Materials and Methods

### 4.1. Detection of Genomic Sequences Identical or Highly Similar to pfeRNAs, and Their Chromosomal Location for Pulmonary Nodule-Related pfeRNAs

We searched for genomic sequences identical or highly similar to published human pfeRNA sequences using the GenBank database (National Library of Medicine) and the BLASTN program (nucleotide collection [nr/nt]; expect threshold: 0.5; mismatch scores: 2, -3; gap costs: linear; up to 5000 sequences) [113], which evaluates sense and antisense strands. For the matches shown, the Expect values, describing the number of hits one can expect by chance when searching a database of a particular size, are less than e^−4^. A first search was performed without organism restriction and a subsequent one specifying “Homo sapiens” as organism.

Chromosomal locations and other details of the detected genomic sequences were obtained from the GenBank database annotations. Chromosomal locations for the pulmonary nodule-related pfeRNAs had been predicted using the QIAGEN CLC Genomics Workbench 10.11 software package [18].

### 4.2. Visualization of RNA Secondary Structures and Estimation of Their Minimum Free Energies

RNA secondary structures were visualized using forna, a force directed graph layout (ViennaRNA Web services) [114]. Optimal secondary structures were also visualized using the RNAfold webserver, which was used to estimate the minimum free energy reflecting the robustness of the pairings [115,116].

## 5. Conclusions

Beyond the function of messenger RNA as a template for protein synthesis, characterizing pfeRNAs enriches the central molecular biology dogma by highlighting how noncoding RNAs directly bind to and regulate the function of intracellular and surface proteins. The findings of the current analysis underscore the concept of a gene capable of producing not only one product, in this case an RNA, but multiple ones, including the most abundant transfer and ribosomal RNAs and their fragments. This is exemplified by the various noncoding RNA fragments generated, often in response to cellular stress, which have diverse functions that enrich the targetome and facilitate the tracking of biogenesis to gene expression, rather than degradation pathways. Conversely, the same RNA fragment can have a multigene origin. The chromosomal location of these genes is diverse, including mitochondrial and nuclear genomes, introns, gene clusters, and genes encoding messenger, noncoding, or multifunctional messenger and noncoding RNAs.

Other known levels of versatility in RNA genes and the functions of their transcripts include bi- or dual-function RNAs [117,118], such as mRNAs that encode noncoding RNAs and, conversely, noncoding RNAs that encode proteins [119]. The bifunctional transfer-messenger RNA (tmRNA) exhibits properties of both tRNA and mRNA, facilitating trans-translation by releasing ribosomes stalled during translation and targeting nascent polypeptides for degradation. This concerted reaction contributes to translational quality control and regulation of bacterial gene expression. Underscoring its importance for bacterial fitness, tmRNA is conserved and one of the most abundant RNAs among bacteria [120].

Additional examples of bifunctional RNAs include the steroid receptor activator/SRA [121], VegT RNA [122,123], Oskar RNA [124], ENOD40 [125], p53 RNA [126], SR1 RNA [127], and Spot 42 RNA [128,129]. As further exemplified here by the matches between pfeRNAS and other ncRNAs, many ncRNAs encompass different ncRNA categories, for instance, H/ACA box small nucleolar (sno)RNA and microRNA [130,131].

Insights into the versatility of gene expression, posttranscriptional modifications, and the complex regulatory network of noncoding RNAs, which account for most of the transcriptome, will continue to enrich the diagnostic, prognostic, and therapeutic armamentarium against human diseases, with applications in all kingdoms of life. It remains to be determined if pfeRNAs originate from their genes, differentially modified fragments of various noncoding RNAs, or both. The biogenesis of pfeRNAs has not been studied, and this analysis opens up the possibility that pfeRNAs could be derived from fragments of known ncRNAs that undergo differential modification.

Table 1 summarizes the characteristics of pfeRNAs relative to well-characterized ncRNAs.

The table highlights the contribution of this manuscript to determining matches between pfeRNAs and fragments of other ncRNAs that were not previously observed, likely secondary to limitations of the commercialized algorithms used. The term pfeRNA was introduced after the first pfeRNA was characterized and termed a piRNA. pfeRNAs have so far been identified in somatic cells, while piRNAs are mainly present in the germline. Although some plant piRNAs have a 2′-O-methylated 3′end, all pfeRNAs thus far characterized are from humans. Further pfeRNA studies on biogenesis based on the present observations and across different organisms are warranted.

The identities described here could form the basis of therapeutic interventions once synergistic and antagonistic properties of the differentially modified ncRNA fragments are further studied.

## Figures and Tables

**Figure 1 ijms-26-06870-f001:**
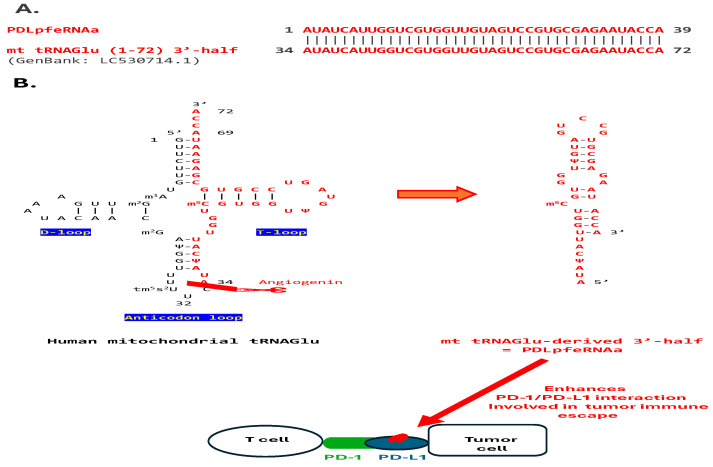
**Sequence analysis of PDLpfeRNAa.** (**A**). PDLpfeRNAa matches the 3′ half of human mt tRNAGlu. (**B**). Angiogenin can cut the mt tRNAGlu at the site depicted by a pair of scissors, and the 3′-half whose secondary structure is visualized might, as PDLpfeRNAa, bind to PD-L1.

**Figure 2 ijms-26-06870-f002:**
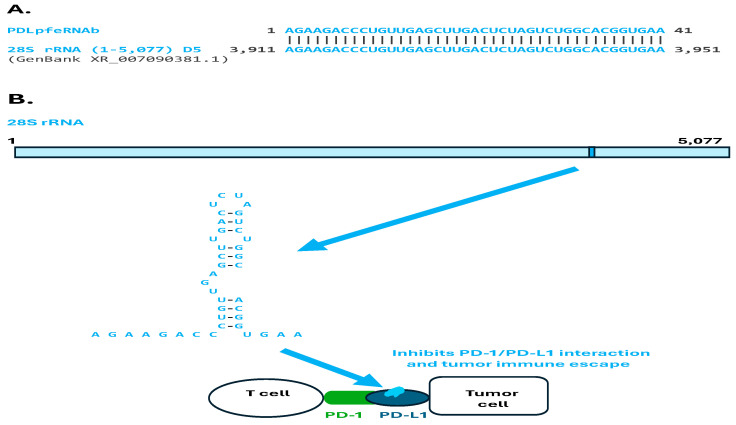
**Sequence analysis of PDLpfeRNAb**. (**A**). PDLpfeRNAb matches a fragment located towards the 3′ end of human 28S rRNA. (**B**). The 28S rRNA fragment whose secondary structure is visualized might, as PDLpfeRNAb, bind to PD-L1.

**Figure 3 ijms-26-06870-f003:**
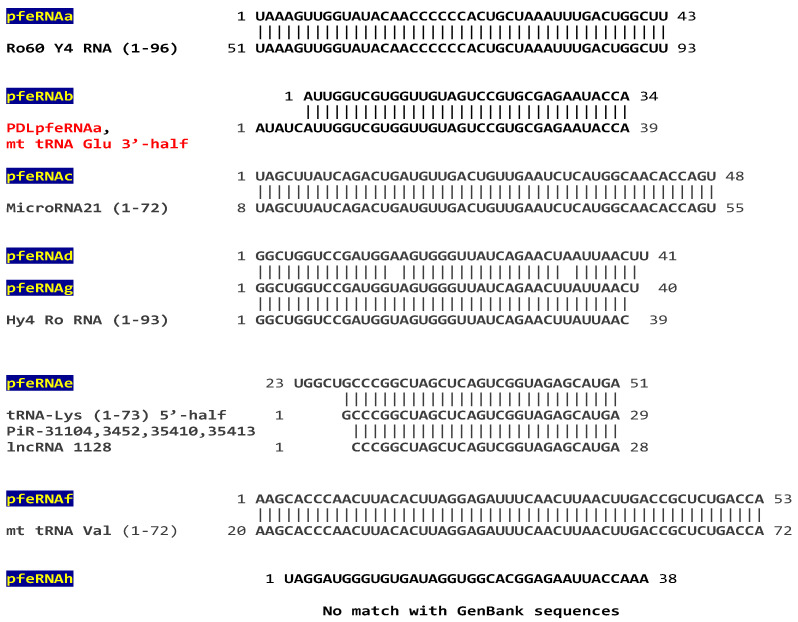
**Sequence analysis of the eight plasma pfeRNAs (a-h) related to pulmonary nodule presence and nature.** pfeRNAb overlaps with PDLpfeRNAa (in red as in Figure 1) except for five nucleotides. pfeRNAs d and g also overlap, differing by three nucleotides.

**Figure 4 ijms-26-06870-f004:**
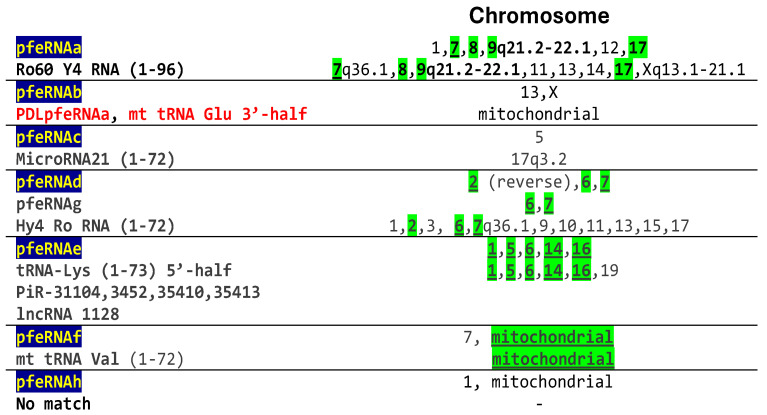
Chromosomal location of pulmonary nodule-related pfeRNAs and matching GenBank sequences shown in Figure 3. Overlapping locations are highlighted in green.

**Figure 5 ijms-26-06870-f005:**
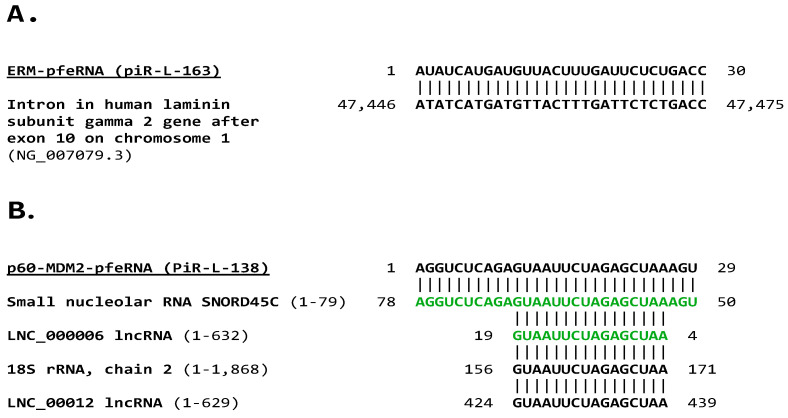
Sequence analysis of pfeRNAs that bind to (**A**). ERM proteins and (**B**). p60-MDM2. Complementary sequence matches are shown in green and reverse numbering corresponding to sense-strand positions.

**Table 1 ijms-26-06870-t001:** Comparison between pfeRNAs and other ncRNAs (based on refs. [12,13,14,15,16,17,18]).

Feature	pfeRNAs	miRNAs, siRNAs, piRNAs, and Other ncRNAs
**Length**	26–60 nts; longer than miRNAs, siRNAs, piRNAs, but shorter than pre-miRNAs, snoRNAs, snRNAs, and others	miRNAS and siRNAs: ~21–23 nts; piRNAs: ~24–31 nts; pre-miRNAs, snoRNAs, snRNAs: >60 nts
**Database presence**	According to previous publications: Not found in existing RNA databases; not predicted by current RNA prediction software. This publication: Matches in GenBank database with fragments of other ncRNAs	Well-annotated in databases like miRbase, piRNABank, etc.
**3′ end** **modification**	2′-O-methylation at the 3′ end, conferring stability against RNase degradation	miRNAs: some have 2′-O-methylation (mainly in plants); piRNAs: 2′-O-methylation; others vary
**Cellular origin**	Identified in somatic cells (not restricted to germline)	piRNAs: mainly germline; miRNAs, siRNAs: somatic and germline
**Target interaction**	Directly bind to specific proteins; regulate protein function without altering protein levels	miRNAs/siRNAs: bind to mRNA targets, regulate gene expression via degradation or translation inhibition; piRNAs: transposon silencing
**Target specificity**	One protein can be affected by more than one pfeRNA; a pfeRNA may recognize proteins with common phosphorylated sites	miRNAs/siRNAs: sequence-specific to mRNA; piRNAs: sequence-specific to transposons
**Functional role**	Modulate biological activities of target proteins (e.g., cell growth, immune escape)	miRNAs/siRNAs: posttranscriptional gene regulation; piRNAs: genome defense
**Biogenesis &** **discovery**	Not well understood; discovered via protein immunoprecipitation and deep sequencing	Biogenesis pathways are well-characterized for most other ncRNAs and their fragments

## Data Availability

All new data created and datasets analyzed are specified in the manuscript text.

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
