# Peer review of "Protein Functional Effector (pfe) Noncoding RNAS Are Identical to Fragments from Various Noncoding RNAs"

_ijms, 2025, doi:10.3390/ijms26146870_

Round 1
Reviewer 1 Report
Comments and Suggestions for Authors
This manuscript explores the identity and function of protein functional effector RNAs (pfeRNAs) in human biology, particularly their sequence similarity to annotated noncoding RNAs (ncRNAs). The study provides a comprehensive computational analysis showing that many pfeRNAs previously identified in cancer biology correspond to fragments of various ncRNAs (e.g., tRNAs, rRNAs, Y RNAs, microRNAs, snoRNAs, lncRNAs). The authors argue for their functional relevance in regulatory mechanisms, tumor immune escape, and potential extracellular signaling, emphasizing implications for diagnostics and therapeutics.
The manuscript makes a compelling case for further exploration of these pfeRNAs. The connections drawn between structural features, sequence identity, and biological function are intellectually stimulating. However, several clarifications, improvements, and refinements are necessary before publication.
Major Comments
- Novelty and Significance
- Comments: The cross-referencing of known pfeRNAs with GenBank-annotated ncRNA fragments offers new insight into their likely origin, multifunctionality, and regulatory potential. It also challenges the idea that these small RNAs are independent entities.
- Suggestion: While the manuscript cites substantial prior work by the authors on pfeRNAs, more explicit comparisons to other small ncRNA fragments (e.g., tRFs, sno-derived RNAs, rRFs, piRNAs) would be helpful to distinguish what makes pfeRNAs unique or functionally distinct.
- Suggestion: Add a comparative table summarizing key distinguishing features between pfeRNAs and other similar small ncRNAs (e.g., modification patterns, biogenesis, known functions, protein interactions).
Abstract
- Well-structured and informative.
- Comment: The abstract is a bit dense; consider breaking up complex ideas (e.g., multifunctionality, synergistic effects) into shorter sentences to enhance readability.
Introduction
- Comment: Some definitions could benefit from simplification. For example, the description of pfeRNAs should be clearer on their functional uniqueness (direct protein interaction).
- Suggestion: Clarify early that the main objective of this paper is the demonstration that known pfeRNAs match known ncRNA fragments and discuss the significance of this finding in the context of ncRNA research.
Results (Sections 2.1–2.3)
- Strength: Detailed sequence alignments and structural analyses are impressive. Figures are informative and clear.
- Concern: Some conclusions (e.g., functional significance of overlapping sequences) rely on inference rather than direct evidence.
Suggestions:
- Validation – Consider discussing whether functional equivalence between matched fragments and known ncRNAs has been experimentally tested or remains speculative.
- Figures 1–5 – The legends could better emphasize functional implications (e.g., what the secondary structure suggests about binding potential).
- Chromosomal Location (2.2) – Clarify the discrepancy between the GenBank-assigned locations and those predicted with QIAGEN software. Could these differences be due to nuclear mitochondrial DNA sequences (NUMTs)?
Discussion
- At times, the discussion reads like a literature review without clearly distinguishing between what was newly discovered in this paper versus what is known.
Suggestions:
- Use subheadings to clearly delineate the study's original findings (sequence matches) vs. contextual background (e.g., isomiRs, glycoRNAs, sno-derived RNAs).
- Provide a schematic model summarizing the proposed role of pfeRNAs across multiple cellular compartments (nucleus, cytoplasm, extracellular).
Conclusion
- Suggestion: Consider adding a paragraph on future directions, especially whether synthetic or engineered pfeRNAs could be used therapeutically.
Minor Issues and Technical Corrections
- Line numbering: Consider adding line numbers in the manuscript for easier reference during peer review.
- Typographical issues:
- “favoring or inhibiting tumor immune escape” – redundant phrasing, revise for clarity.
- “not-of-degradation-or-precursor-maturation-origin” – rephrase for clarity (e.g., "not derived from degradation or precursor maturation").
- Reference style: Ensure consistency in citation format (some are missing journal names or DOIs).

Author Response
Major Comments
- Novelty and Significance
- Comments: The cross-referencing of known pfeRNAs with GenBank-annotated ncRNA fragments offers new insight into their likely origin, multifunctionality, and regulatory potential. It also challenges the idea that these small RNAs are independent entities.
- Suggestion: While the manuscript cites substantial prior work by the authors on pfeRNAs, more explicit comparisons to other small ncRNA fragments (e.g., tRFs, sno-derived RNAs, rRFs, piRNAs) would be helpful to distinguish what makes pfeRNAs unique or functionally distinct.
- Suggestion: Add a comparative table summarizing key distinguishing features between pfeRNAs and other similar small ncRNAs (e.g., modification patterns, biogenesis, known functions, protein interactions).
We thank the reviewer for the thorough and valuable review. All changes are marked in red letters throughout the manuscript.
We added the recommended Table to the end of the discussion. The Table summarizes what is known about both pfeRNAs and other ncRNAs. We also followed the recommendations below to emphasize the distinction between what is known and what is new, as outlined in this paper. To this end, we also added ‘has been shown to’ and ‘here in multiple locations.’ We also added subsections of what is known as recommended.
As mentioned in several sections, the findings of this paper should encourage further research by groups that focus on the different ncRNAs.
Abstract
- Well-structured and informative.
- Comment: The abstract is a bit dense; consider breaking up complex ideas (e.g., multifunctionality, synergistic effects) into shorter sentences to enhance readability.
We have modified the Abstract accordingly while respecting the 250-word limit.
Introduction
- Comment: Some definitions could benefit from simplification. For example, the description of pfeRNAs should be clearer on their functional uniqueness (direct protein interaction).
- Suggestion: Clarify early that the main objective of this paper is the demonstration that known pfeRNAs match known ncRNA fragments and discuss the significance of this finding in the context of ncRNA research.
We have now highlighted the differentiating features of pfeRNAs in various sections (including the addition of a Table) and discussed how the paper's findings provide a valuable basis for further research, not only on pfeRNAs but also on all ncRNAs. The fact that the identities described were missed for a decade also highlights the need to test the uniqueness of newly characterized RNA sequences more rigorously. As shown, this does not necessitate the development of new algorithms. It suggests that the completeness of databases and the modification of commercially available software would be useful.
Results (Sections 2.1–2.3)
- Strength: Detailed sequence alignments and structural analyses are impressive. Figures are informative and clear.
- Concern: Some conclusions (e.g., functional significance of overlapping sequences) rely on inference rather than direct evidence.
Suggestions:
- Validation – Consider discussing whether functional equivalence between matched fragments and known ncRNAs has been experimentally tested or remains speculative.
- Figures 1–5 – The legends could better emphasize functional implications (e.g., what the secondary structure suggests about binding potential).
- Chromosomal Location (2.2) – Clarify the discrepancy between the GenBank-assigned locations and those predicted with QIAGEN software. Could these differences be due to nuclear mitochondrial DNA sequences (NUMTs)?
We have added the following statement:
We have added the following statement: The chromosomal locations of these pfeRNAs need to be determined not through prediction, but rather through a more direct methodology, as is the case for the annotated sequences to which they match here.
Discussion
- At times, the discussion reads like a literature review without clearly distinguishing between what was newly discovered in this paper versus what is known.
Suggestions:
- Use subheadings to clearly delineate the study's original findings (sequence matches) vs. contextual background (e.g., isomiRs, glycoRNAs, sno-derived RNAs).
We have added subheadings.
- Provide a schematic model summarizing the proposed role of pfeRNAs across multiple cellular compartments (nucleus, cytoplasm, extracellular).
We have mentioned throughout the text that differential modifications might allow pfeRNAs to exert functions in different compartments, as has been shown for the matching ncRNAs.
Conclusion
- Suggestion: Consider adding a paragraph on future directions, especially whether synthetic or engineered pfeRNAs could be used therapeutically.
We added the statement at the end: The identities described here could form the basis of therapeutic interventions once synergistic and antagonistic properties of the differentially modified ncRNA fragments are further studied.
Minor Issues and Technical Corrections
- Line numbering: Consider adding line numbers in the manuscript for easier reference during peer review.
- Typographical issues:
- “favoring or inhibiting tumor immune escape” – redundant phrasing, revise for clarity.
- “not-of-degradation-or-precursor-maturation-origin” – rephrase for clarity (e.g., "not derived from degradation or precursor maturation").
- Reference style: Ensure consistency in citation format (some are missing journal names or DOIs).
We have corrected these issues. Thank you for your detailed review. In terms of line numbering, the version we sent and the revised one both have line numbering. We also checked the references and they all have journal names and when available DOIs. We apologize if you were sent a different version.
Reviewer 2 Report
Comments and Suggestions for Authors
This manuscript demonstrates that many of pfeRNAs are identical or highly similar to annotated fragments of human ncRNAs through sequence alignment. The authors hypothesize that these pfeRNAs may exert regulatory functions. However, the manuscript is primarily a survey based on sequence similarity, without offering novel methodological contributions or functional validation.
Comments:
- The study is built entirely on standard BLAST sequence alignment without developing new computational tools, statistical models, or analytical frameworks. This level of computational work falls below the typical threshold of novelty for publications in IJMS.
- No experimental validation is presented to support the functional claims made about the identified pfeRNAs. While the manuscript cites previous studies, it does not test any of the hypothesized regulatory roles, protein interactions, or disease relevance of the pfeRNAs.
Author Response
- The study is built entirely on standard BLAST sequence alignment without developing new computational tools, statistical models, or analytical frameworks. This level of computational work falls below the typical threshold of novelty for publications in IJMS.
- No experimental validation is presented to support the functional claims made about the identified pfeRNAs. While the manuscript cites previous studies, it does not test any of the hypothesized regulatory roles, protein interactions, or disease relevance of the pfeRNAs.
We thank the reviewer for the two comments, which have helped improve the content presentation in the manuscript. We apologize for not presenting our findings more clearly initially. We now distinguish more explicitly what was known about pfeRNAs and what our analysis revealed.
Yes, it was simply a matter of using the standard BLAST sequence alignment tool to demonstrate that most known pfeRNAs are not novel in primary structure, as had been claimed for a decade in all published studies on their detection, function and clinical utility.
The fact that this identity was overlooked over the years warrants caution in relying solely on commercially provided methods for analyzing sequences for redundancy with known ones. We did not specify this too bluntly because we highly respect and applaud the work of the investigators on pfeRNAs. We also did not intend to demean commercially developed tools openly, but rather to suggest tacitly that they can be improved.
pfeRNAs have been distinguished experimentally from other ncRNAs by their 3’-end modification, namely 2’-O-methylation, and by their exerting their functions by directly binding proteins but not altering their levels. The observations in the paper, therefore, show that known so-called pfeRNAs can be derivved from different known ncRNA genes via differential modifications, thereby enriching the known multifunctionality of fragments derived from them, including ones that have been studied for decades, such as transfer and ribosomal RNAs.
We have modified all sections of the manuscript to address the reviewer’s comments (marked in red letters and including a Table) and hope that they will be to the reviewer’s satisfaction.
Round 2
Reviewer 2 Report
Comments and Suggestions for Authors
The authors have improved their manuscript significantly with more clarification and a summary table.